# Regulation of IGF1R by MicroRNA-15b Contributes to the Anticancer Effects of Calorie Restriction in a Murine C3-TAg Model of Triple-Negative Breast Cancer

**DOI:** 10.3390/cancers15174320

**Published:** 2023-08-29

**Authors:** Ximena Bustamante-Marin, Kaylyn L. Devlin, Shannon B. McDonell, Om Dave, Jenna L. Merlino, Emma J. Grindstaff, Alyssa N. Ho, Erika T. Rezeli, Michael F. Coleman, Stephen D. Hursting

**Affiliations:** 1Department of Nutrition, University of North Carolina, Chapel Hill, NC 27599, USA; 2Nutrition Research Institute, University of North Carolina, Chapel Hill, NC 28081, USA; 3School of Medicine, Oregon Health and Science University, Portland, OR 97239, USA; devlink@ohsu.edu; 4Lineberger Comprehensive Cancer Center, University of North Carolina, Chapel Hill, NC 27599, USA

**Keywords:** calorie restriction, miR-15b, IGF1R signaling, triple-negative breast cancer, breast cancer

## Abstract

**Simple Summary:**

Breast cancer is the most common cancer in women worldwide. The risk of developing postmenopausal breast cancer is exacerbated by obesity (BMI > 30 kg/m^2^), and growing evidence suggests that obesity increases the risk of developing triple-negative breast cancer. Calorie restriction (CR), a reduction of calorie intake by 20–40% without causing malnutrition, is associated with decreasing an individual’s risk of developing cancer, enhancing the responses to cancer treatment, and reducing the risk of breast cancer recurrence. The negative impact of CR on breast cancer growth may result from lowering bioavailable levels of IGF1. In this study, we suggest that CR’s antitumor effects are partly mediated by the upregulation of miR-15b, which downregulates IGF1R and other target genes involved in cell cycle control. Our findings suggest that miR-15b could mediate CR’s regulation of IGF1/IGF1R signaling that contributes to the anticancer properties of this dietary intervention.

**Abstract:**

Calorie restriction (CR) inhibits triple-negative breast cancer (TNBC) progression in several preclinical models in association with decreased insulin-like growth factor 1 (IGF1) signaling. To investigate the impact of CR on microRNAs (miRs) that target the IGF1/IGF1R pathway, we used the spontaneous murine model of TNBC, C3(1)/SV40 T-antigen (C3-TAg). In C3-TAg mice, CR reduced body weight, IGF1 levels, and TNBC progression. We evaluated the tumoral expression of 10 miRs. CR increased the expression of miR-199a-3p, miR-199a-5p, miR-486, and miR-15b. However, only miR-15b expression correlated with tumorigenicity in the M28, M6, and M6C C3-TAg cell lines of TNBC progression. Overexpressing miR-15b reduced the proliferation of mouse (M6) and human (MDA-MB-231) cell lines. Serum restriction alone or in combination with low levels of recombinant IGF1 significantly upregulated miR-15b expression and reduced *Igf1r* in M6 cells. These effects were reversed by the pharmacological inhibition of IGFR with BMS754807. In silico analysis using miR web tools predicted that miR-15b targets genes associated with IGF1/mTOR pathways and the cell cycle. Our findings suggest that CR in association with reduced IGF1 levels could upregulate miR-15b to downregulate *Igf1r* and contribute to the anticancer effects of CR. Thus, miR-15b may be a therapeutic target for mimicking the beneficial effects of CR against TNBC.

## 1. Introduction

Breast cancer is the most common cancer in women worldwide [1]. Unfortunately, the risk of developing postmenopausal breast cancer has been exacerbated by the worldwide rise in obesity (BMI ≥ 30 kg/m^2^). Growing evidence suggests that abdominal obesity increases the risk of developing triple-negative breast cancer (TNBC), which is more common in premenopausal women [2]. Population studies have shown that for every 20% increment in calorie intake, the risk of developing TNBC increases by 7% [3]. In addition, the obesity-related metabolic perturbations, including dysregulation of insulin, IGF1, adipokines, inflammatory factors (e.g., cytokines), and vascular integrity-related factors (e.g., vascular endothelial growth factor) [4] promote a tumor microenvironment associated with a reduced efficacy of anticancer treatments [5,6]. Thus, identifying nutritional and lifestyle changes that can modify breast cancer risk and improve treatment outcomes are urgently needed [5,6].

Calorie restriction (CR) involves reducing calorie intake by 20–40% without causing malnutrition. This dietary modification is associated with increased longevity and has been investigated as a treatment option for various age-related diseases [6]. CR decreases an individual’s risk of developing cancer, enhances the responses to cancer treatment [6], and reduces the risk of recurrence for breast cancer survivors [7]. The prohibitive effects of CR on breast cancer growth may result from the lowering effects of CR on bioavailable levels of insulin, IGF1, leptin and pro-inflammatory molecules; thus, CR impacts molecular pathways leading to the suppression of cell growth and angiogenesis.

Although the beneficial properties of CR have been extensively demonstrated [7,8,9], due to the broad-acting nature of this dietary intervention, the systemic and molecular mechanisms responsible for the observed effects have not been fully explained. It is clear, however, that the regulation of hormonal and adipose-derived factors (insulin, IGF1, and leptin) plays a major role in the anticancer effects of CR. In particular, bioavailable IGF1 has emerged as a key mitogen modulated by CR [9,10]. IGF1 impacts local and systemic growth and survival by activating the phosphatidylinositol-3 kinase (PI3K)/Akt pathway through IGF1 receptor (IGF1R) binding. The PI3K/Akt pathway integrates intracellular and environmental cues regarding nutrient availability to regulate cellular proliferation, survival, and protein translation through several downstream mediators, including the mammalian target of rapamycin (mTOR) [11,12]. The IGF1R is commonly expressed in human tumors leading to a mitogenic response to physiological concentrations of IGF1, and both the PI3K/Akt pathway and mTOR are commonly activated in cancers [13]. Prospective analyses suggest an association between circulating IGF1 and breast cancer risk [14,15]. Also, the gene expression pattern induced by IGF1 predicts poor outcomes in patients with breast cancer [16].

MicroRNAs (miRs) are small noncoding RNA molecules that regulate gene expression by promoting mRNA degradation and/or repression of translation. Several IGF1R/PI3K/Akt/mTOR pathway components are regulated by miRs, for example, IGF1R (miR-15b, -16, -99a, -122, -486), insulin receptor substrate-1 and -2 (miR-7, -145, -148a, let-7), Akt (miR-100), mTOR (miR-99, -100, -101, -199), and PTEN (miR-21, -26, -19, -221) [17,18]. The functions of many of these miRs in relation to their targets were discovered due to their dysregulation in cancers. Recent evidence has shown that miRs may be critical mediators of CR’s beneficial effects on metabolism and physiology [19], including carcinogenesis [20,21], although their role in the anticancer effects of CR on TNBC has not been elucidated. We therefore evaluated the impact of CR on known miRs that target the IGF1R/PI3K/Akt/mTOR pathway components in a C3(1)/SV40 Tumor-Antigen (C3-TAg) transgenic mouse model of spontaneous basal-like TNBC [22]. We identified several miRs upregulated in tumors from CR mice, however, only miR-15b expression was affected by the tumorigenicity of the C3-TAg series of progressively aggressive mammary tumor cells. In this study, we hypothesized that the antitumor effects of CR are in part mediated by the upregulation of miR-15b via its regulatory effect on *Igf1r* expression. Our results delineate a possible mechanism in which CR results in increased levels of miRNA-15b that contribute to the downregulation of IGF1R and other target genes involved in cell cycle regulation. Thus, our findings suggest that miR-15b is another component of CR’s regulation of IGF1 signaling that may contribute to the anticancer properties of this dietary intervention.

## 2. Materials and Methods

### 2.1. Animal Study Design

All animal studies were performed with the approval of the University of North Carolina at Chapel Hill Institutional Animal Care and Use Committee. All diets were purchased from Research Diets, Inc., New Brunswick, NJ. C3-TAg transgenic mice on an FVB/N background were bred in-house in collaboration with UNC Lineberger Comprehensive Cancer Center’s Mouse Phase I Unit. This model closely mimics human basal-like TNBC due to its lack of estrogen receptor alpha, progesterone receptor, and human epidermal growth factor 2 expression [22]. C3-TAg female mice develop atypical ductal hyperplasia beginning at 8 weeks of age, then these lesions progress to mammary intraepithelial neoplasia (MIN) confined within the basement membrane of the duct, which are histologically similar to human ductal carcinoma in situ, within 12 weeks of age. Finally, invasive ductal carcinoma develops, often multifocally, within several mammary fat pads, around 16 weeks of age [22]. Female mice (*n* = 140) were weaned at three weeks of age, fed chow diet for one week, then singly housed and randomized to receive either a modified AIN-93M control diet consumed ad libitum (*n* = 66; catalog #D12450B) or a CR diet regimen (*n* = 74). CR mice received a modified formulation of the control diet (catalog #D03020702). To adjust the amount of food that the mice ate as they aged and grew, each CR mouse was fed a daily food aliquot that contained 70% of the average calories consumed the previous week by control mice of the same age such that daily aliquots of food provided 70% of the mean daily calories (and 100% of the vitamins, minerals, fatty acids and amino acids). Any food from that aliquot not consumed the next day was collected, measured and recorded, although this was very rare as the mice almost always consumed the entire aliquot. Water was provided ad libitum. Mice were further randomized into 2 substudies: (A) Time course study, in which 8 mice/diet/time point were euthanized at 8, 11, 14, 17, 20 weeks of age and mammary fat pads 2, 4, 7, and 9 were excised and processed for histopathology analysis. Because the CR regime slows down tumor growth, a 23-week time point was also included for CR mice to increase the likelihood of detecting all grades of tumor lesion. (B) Survival study, in which female mice (*n* = 26/diet) were palpated twice weekly until a primary tumor reached 1.2 cm in any direction. The mouse was euthanized and tumors were collected. In addition, from this mouse study, 3 mice/diet were euthanized at 5 and 8 weeks of age for mammary fat pad whole-mount preparation to assess preneoplastic lesion formation. Two control mice and one CR mouse were censored from data before tumor detection and one control mouse and four CR mice were censored from data after tumor detection but before maximum tumor size was reached, all due to health issues or death from non-mammary tumor-related causes. All mice were monitored and weighed daily. Control food was measured and replenished once per week, while the CR group was administered their food in daily aliquots. Following euthanasia by CO_2_ inhalation, blood was collected by cardiac puncture and allowed to coagulate for 45 min at room temperature. Serum was obtained following a serial centrifugation at 1500× *g* for 15 min, and 10,000× *g* for 10 min, and stored at −80 °C.

### 2.2. Assesment of Tumor Development Assessment

To evaluate tumor development, mammary fat pads were palpated twice weekly during the study. All tumors discovered by palpation were measured in 2 dimensions using electronic calipers until the tumor reached 1.2 cm in any dimension, at which point the mouse was euthanized and tissues were collected. Secondary tumors were defined as any tumor that was not the determining tumor (primary tumor) for euthanasia based on tumor size. There were no differences in size of the primary tumors between the two diet groups (data not shown) because they were allowed to grow to a pre-specified maximum size (1.2 cm in any direction) for survival analysis.

### 2.3. Tissue Processing and Histopathology

Whole mammary fat pads harvested from the time course study were fixed overnight in methacarn fixative (to help preserve nucleic acid integrity), embedded in paraffin, and cut into 4 μm sections. Slides were stained with H&E and histologically assessed (in a blinded manner) by a board-certified veterinary pathologist into one of three pre-malignant stages of cancer progression: normal tissue, atypical hyperplasia (AH), or mammary intraepithelial neoplasia (MIN). Primary tumors from the survival study were split longitudinally, one half being snap-frozen in liquid nitrogen and stored at −80 °C and one half being fixed in 10% neutral-buffered formalin for 24 h, transferred to 70% ethanol for at least 24 h, and embedded in paraffin. All secondary tumors from the survival study were snap-frozen only.

### 2.4. Serum Hormone Analysis

Serum insulin and leptin concentrations were measured using the Bio-Plex Pro mouse diabetes panel (BioRad, Hercules, CA, USA). Serum IGF1 concentrations were measured using the IGF1 Mouse Magnetic Luminex Screening Assay (R&D Systems, Minneapolis, MN, USA). Assays were conducted according to manufacturer’s protocols on a Bio-Plex MAGPIX system (BioRad).

### 2.5. Cell Culture

The C3-TAg progression series of cell lines M28 (weakly tumorigenic), M6 (invasive primary carcinoma), and M6C (aggressive, metastatic) were a kind gift from Dr. Jeffery Green at the National Cancer Institute [23]. MDA-MB-231 (human metastatic breast cancer cell line) were acquired from ATCC (Gaithersburg, MD, USA). C3-TAg cell lines were grown in complete DMEM (25 mM glucose), and MDA-MB-231 cells were grown in complete RPMI (11 mM glucose), each supplemented with 10% FBS, 2mM glutamine, and 100 U/mL penicillin–streptomycin unless otherwise indicated. For IGF1 treatment, 3 × 10^5^ cells were seeded into 6 cm plates and cultured in complete media (CM, described above) for 12 h, before being cultured for 4 h in media without FBS. Subsequently, the cells were cultured for 18 h in media without FBS, supplemented with 10 or 30 ng/mL of IGF1. To emulate CR in vitro, 3 × 10^5^ M6 cells were seeded into 6 cm plates and cultured for 12 h in CM, before being cultured for 4 h in serum-free media (SFM) and subsequently treated with one of the following conditions for 18 h: CM; CM + 100 nM BMS754807 (MedChemExpress, inhibitor of IGF1R); SFM + 0.5% FBS; SFM + 0.5% FBS + 10 ng/mL IGF1; SFM + 0.5% FBS + 100 nM BMS754807; and SFM + 0.5% FBS + 10 ng/mL IGF1 + 100 nM BMS754807. Cells were maintained under 5% CO_2_ at 37 °C, and all reported data were acquired using cells between passage number 12 and 20.

### 2.6. Western Blot

Tumor and cell line protein lysates were extracted using RIPA buffer supplemented with protease Inhibitor Cocktail tablets (Sigma-Aldrich, St. Louis, MO, USA), Phosphatase Inhibitor Cocktail 2 (Sigma-Aldrich), and Phosphatase Inhibitor Cocktail 3 (Sigma-Aldrich). Total protein concentration was determined using Bio-Rad Protein Assay Dye Reagent (Bio-Rad) with bovine serum albumin as standards. Protein samples were resolved by SDS-PAGE on 12% acrylamide gels (BioRad) and transferred to PVDF membranes (Bio-Rad). After blocking membranes were probed with antibodies specific to IGF1R (1:1000, #9750 and #3027, Cell Signaling, Danvers, MA, USA), Tubulin (1:5000, ab6046 Abcam, Cambridge, UK), Actin (1:5000, Sigma-Aldrich), and the phospho-ribosomal protein S6 (p-RPS6) (1:1000, #2211s Cell Signaling). Membranes of tumor samples were developed using HRP-conjugated secondary antibodies (GE Healthcare Life Sciences, Pittsburgh, PA, USA) and chemiluminescence (Amersham ECL Prime Western Blotting Detection Reagent). The membranes of cell line samples were developed using fluorescent Western blot detection (LI-COR, Lincon, NE, USA). Relative protein levels were determined after normalization with tubulin or Actin.

### 2.7. RNA Extraction, miR and Gene Expression Analysis

Secondary mammary tumors from mice in the control and CR diet (*n* = 5–7/diet group) groups in the survival study were collected and total RNA was extracted using TRI Reagent^®^ according to the manufacturer’s instructions. The total RNA from cell lines was isolated using PureLink RNA Mini Kit (Thermo Fisher Scientific, Walthman, MA, USA). RNA quantity was measured on a Nanodrop 2000 spectrophotometer. Complementary DNA of miRs was generated using the Universal cDNA Synthesis Kit II (Exiqon, Vedbaek, Denmark) or TaqMan™ Advanced miRNA cDNA Synthesis Kit (Thermo Fisher). The expression of miR-15b was analyzed using TaqMan Universal master Mix II, no UNG and TaqMan assay (Thermo Fisher Scientific) or using LNA PCR primer sets and ExiLENT SYBR Green master mix (Exiqon). The expression of miR-15b target genes was evaluated using specific primers (Appendix A) with SyBR Green master mix (BioRAD) on a ViiA™ 7 Real-Time PCR system (Applied Biosystems, Carlsbad, CA, USA). All steps of assays were performed according to the manufacturer’s protocols. miR-16-5p and *Actb* were used as reference genes for PCR-based analyses. miR-16-5p was selected as an endogenous control based on available literature at the time of the study [24,25]. We evaluated the expression of mir-16 in terms of Ct value under the multiple conditions analyzed and we did not observe significant differences. Relative expression calculations were carried out using the ΔΔCt method.

### 2.8. Luciferase Assay and miR-15b Overexpression

The cell lines MDA-MB-231 and M6 were co-transfected in tandem with 10 nM miR-15b mimic or control mimic (Exiqon) and a synthetic miR-15b target RenSP luciferase reporter plasmid (Switchgear Genomics, Carlsbad, CA, USA) at a ratio of 1:5 (ng:nL) with Lipofectamine 3000 transfection reagent (Thermo Fisher Scientific). Twenty-four hours after transfection, cells were seeded into two separate 96-well plates, one for luciferase readout to verify miR-15b overexpression and one for proliferation analysis. Luciferase and proliferation analyses were conducted 48 h after reseeding, for a total of 72 h after miR transfection. RenSP luciferase signal was analyzed using LightSwitch Assay Reagent (Switchgear Genomics, Menlo Park, CA, USA) according to the manufacturer’s protocol. Overexpression of miR-15b was achieved by reverse transfection using Lipofectamine RNAimax (Thermo Fisher Scientific) and 10 nM of miR-15b mimic, which are artificial double-stranded RNAs consisting of the guide strand that mimics the function of the endogenous miRNA or miR-15b inhibitor. The miR-15b inhibitor was designed to be an exact antisense to mir15b forming a duplex with the miRNA-15b guide strand, preventing it from binding to its intended targets and promoting its degradation (Exiqon) [26]. The expression of miR-15b and target genes was evaluated 24 h after transfection.

### 2.9. Cellular Proliferation Assays

Cellular proliferation was measured 48 h after treatment using a BrdU Cell Proliferation ELISA kit (Abcam) according to the manufacturer’s protocols or by measuring cellular metabolic activity, as an indicator of cell viability, proliferation, and cytotoxicity, using the colorimetric assay based on the reduction of yellow tetrazolium salt 3-(4,5-dimethylthiazol-2-yl)-2,5-diphenyltetrazolium bromide (MTT, Sigma-Aldrich), as previously described [27]. The absorbance was read at 570 nm on a Cytation 3 Cell Imaging Multi-Mode Reader (BioTek Instruments Inc., Winooski, VT, USA).

### 2.10. In Silico Analysis of miR-15b Target Prediction

Using miR databases, specifically miRmap, miRDB, miRabel and TargetScan [28,29,30], we generated combined predicted target genes for the two mature forms of human miR-15b (hsa-miR-15b-5p and hsa-miR-15b-3p, Appendix A) and for the two mature forms of murine miR-15b (mmu-miR-15b-5p and mmu-miR-15b-3p, Appendix A). Subsequently, a master list of target genes was generated, in which the predicted target was represented in at least one of the miR databases. To narrow the candidate genes, the master list of predicted targets for human and mouse mir15-b was combined and subjected to further validation in miRTarBase (Appendix A) [31]. Also, we compiled a list of genes involved in the IGF1/mTOR pathway to validate the role of miR-15b as a possible regulator. The common gene targets of mmu-miR-15b and hsa-miR-15b obtained from all miR databases were analyzed through the Database for Annotation, Visualization, and Integrated Discovery (DAVID), facilitating additional functional interpretation of the gene list.

### 2.11. Statistical Analyses

Animal study data are presented as mean ± SD and in vitro data as mean ± SEM. All data shown represent the average of at least 3 independent experiments. Statistical analysis was performed using GraphPad Prism software (GraphPad Software Inc., San Diego, CA, USA), and *p* < 0.05 was considered significantly different. Kaplan–Meier survival curves were plotted, and the between-group difference in survival was analyzed using the log-rank (Mantel–Cox) test. Differences between animals or cells exposed to two different experimental conditions were analyzed using Student’s *t*-test. Differences between cells exposed to more than two experimental conditions were analyzed using one-way ANOVA, followed by Tukey’s post hoc test. Data from experiments with more than one independent variable were analyzed using a mixed-model approach with Geisser–Greenhouse correction, followed by a multiple comparison test using the Holm–Šídák method.

## 3. Results

### 3.1. Protective Effects of CR on Preventing Mammary Tumor Development

**CR promotes long-term weight maintenance and reduces levels of mitogenic hormones.** Several studies have consistently shown that long-term CR without malnutrition is a robust intervention to inhibit cancer and increase life and health span in rodents and humans [9,10,32]. To investigate the impact of CR on mammary cancer we use the C3-TAg transgenic mice that develop spontaneous mammary tumors [22]. Our results show that the CR regimen resulted in an initial loss of body weight of ~2 g followed by long-term body weight maintenance (average 15.1 ± 0.64 g) over 20 weeks (survival and time course study, *p* < 0.0001), relative to mice on a control diet that continuously gained weight until the end of the study (25.1 ± 2.25 g) (Figure 1A).Consistently, CR reduced serum levels of metabolic hormones. A mixed-effect ANOVA revealed that there was a statistically significant interaction between the effects of the diet and the time course of the diet (presented as week of age) on the levels of insulin (F(3,17) = 3.871, *p* = 0.028), leptin (F(4,51) = 2.831, *p* = 0.034), and IGF1 (F(4,54) = 2.631, *p* = 0.044). In addition, simple main-effect analysis showed that CR and time had statistically significant effect on the levels of insulin (*p* = 0.034, and *p* = 0.024, respectively), leptin (*p* < 0.0001, and *p* = 0.020, respectively), and IGF1 (*p* < 0.0001, and *p* = 0.0089). Although the levels of insulin and leptin were reduced in the CR group, this reduction was not statistically significant over the time course, most likely due to the high variation in the measurements. However, the multi-comparison test showed statistically significant reduction in the levels of IGF1 relative to levels in mice under the control diet throughout the 12 weeks of the time-course study (from 8 to 20 weeks of age, Figure 1B). This result is consistent with previous studies in rodents showing a CR-mediated reduction in the levels of IGF1 [9].

**CR protects against tumor development and progression in a mouse model of TNBC.** The CR regimen significantly delayed overall tumor development by 30% compared to the control diet, with a median time to palpable primary tumor of 23 weeks vs. 15 weeks, respectively (Figure 1C; *p* = 0.0002). Thus, the overall survival was increased by 50% in CR (median survival of 30 weeks) relative to the control diet (median survival of 20 weeks, *p* < 0.0001) (Figure 1D). Four mice in the CR group (30 and 34 weeks of age) and two mice on the control diet group (30 and 33 weeks of age) remained tumor free until the end of the survival study. Additionally, the CR regimen reduced the tumor burden by decreasing tumor multiplicity (*p* = 0.025) (Figure 1E) and secondary tumor weight (*p* = 0.050) (Figure 1F) compared with mice on the control diet. Also, in the CR group, relative to the control diet group, tumoral *Igf1r* expression was downregulated by 34% (*p* = 0.030) (Figure 1G) and a modest decrease in tumoral IGF1R protein levels was also observed (Figure 1H, Appendix A). In our time course study, the axillary mammary glands from the control diet and CR mice were evaluated by histopathology every 3 weeks from 8 to 20 weeks of age. In the CR group, the majority of ductal structures remained normal across all time points; only a modest amount of MIN was observed in the glands, and low ductal density within the mammary gland was observed (Appendix A). In contrast, in mice on the control diet, the majority of mammary lesions progressed to MIN by 14 weeks and ductal development toward high density was observed (Appendix A). Together, these results demonstrate the protective effects of CR on preventing mammary tumor development by reducing the levels of mitogenic hormones, tumor burden, decreasing circulating levels of IGF1, and reducing the tumoral mRNA and protein levels of IGF1R; thus, increasing the life span of the C3-Tag mice.

### 3.2. miR-15b a Candidate Mediator of the Beneficial Effects of CR

**CR increases expression of several miRs known to regulate IGF signaling.** We assessed the impact of CR on the tumoral expression of 10 miRs previously reported to target components of the IGF1 and mTOR pathways (Appendix A [33,34,35,36,37,38,39,40,41,42,43,44,45,46,47,48,49,50,51]). Six of the miRs were not differentially expressed (Appendix A), while four were upregulated in secondary tumors from the CR group relative to the control diet: miR-15b (2.19-fold, *p* = 0.018); miR-199a-3p (2.07-fold, *p* = 0.021); miR-199a-5p (1.65-fold, *p* = 0.036); and *miR-486* (6.26-fold, *p* = 0.046) (Figure 2A).

**Expression of miR-15b is affected by cell tumorigenicity.** To investigate the impact of the four upregulated miRs on tumor progression, we evaluated their expression using the cell line series M28, M6, and M6C, derived from C3-TAg mice during different stages of tumor development [23]. In these cell lines, expression of miR-199a-3p and miR-199a-5p was not detected. The expression of miR-15b and miR-486 was reduced as the tumorigenicity of the cell-line progression increased (Figure 2B). In particular, miR-15b was significantly downregulated in metastatic M6C cells compared to M6 (invasive carcinoma) and M28 (weakly tumorigenic) cells. In contrast, miR-486 expression did not differ between M6 and M6C cells. These results suggest that miR-15b downregulation could be associated with increased tumor progression, and therefore we selected miR-15b as a candidate miR for further studies of the anticancer effects of CR in highly proliferative cells.

### 3.3. miR-15b Reduces Cell Proliferation and Targets Igf1r in Mammary Cancer Cell

**miR-15b overexpression inhibits mouse and human TNBC cell proliferation.** To evaluate the impact of miR-15b on cellular proliferation, mouse (M6) and human (MDA-MB- 231) basal-like mammary cancer cell lines were dually transfected with a miR-15b mimic or control mimic in combination with a RenSP luciferase reporter plasmid carrying a synthetic miR-15b target site in the 3′ UTR of the luciferase gene. Seventy-two hours following transfection, increased activity of miR-15b was confirmed with a decreased luciferase signal (M6, *p* = 0.001; MDA-MB-231, *p*= 0.0074) (Figure 3A,B). Cellular proliferation, evaluated by BrdU incorporation, was reduced in both cell lines in response to miR-15b overexpression (M6, 36% reduction, *p* = 0.004; MDA-MB-231, 35% reduction, *p* = 0.002), relative to an miR-15b mimic control (Figure 3C,D). These results highlight the association between increased miR-15b expression and reduction in mammary-tumor-cell proliferation, suggesting a possible antiproliferative effect for miR-15b in mammary tumor cells.

**miR-15b targets *Igf1r* to potentially inhibit mammary cancer cell proliferation.** Studies using mouse models suggest that reduction of bioavailable IGF1 mediates many anticancer effects of CR [52,53,54,55]. Given that we measured increased tumoral expression of miR-15b in CR mice and the reduced tumor expression of *Igf1r*, we interrogated if upregulation of miR-15b affects the levels of IGF1R. To evaluate the impact of miR-15 on *IGF1R* expression, MDA-MB-231, M6 and M6C cells were transfected with an miR-15b mimic to enhance endogenous miR-15 activity, or with a miR-15b inhibitor to block miR-15b binding to its target genes. The transient expression of the miR-15b mimic resulted in the overexpression of miR-15b (Figure 4) in all cells lines; however, the effect on *Igf1r* expression was cell-line-dependent. In MDA-MB-231 (Figure 4A), the overexpression of miR-15b did not have significant effects on the levels on *IGF1R*. We detected a significant reduction of *Igf1r* in M6 cells (Figure 4B) and a slight increase in *Igf1r* in M6C cells (Figure 4C). In contrast, the treatment with an miR-15b hairpin inhibitor blocked miR-15b downregulation of *Igf1r* in all cell lines, causing a significant increase in *Igf1r* (Figure 4A). In addition, in MDA-MB-231 and M6 cells transfected with a miR-15b mimic, we observed a reduction in the protein levels of IGF1R and its downstream effector, the phospho-ribosomal protein S6 (p-RPS6), 48 h after overexpressing miR-15b (Figure 4D, PDF File S1). We did not observe significant changes in the levels of IGF1R and p-RPS6 in M6C cells or in the three cell lines transfected with the miR-15b hairpin inhibitor. Consistent with our results in Figure 3, we measured a mild effect on cellular proliferation at 48 h and a significant reduction at 72 h after miR-15B overexpression in MDA-MB-231 and M6 cells but not in M6C cells. These results suggest that the effects of miR-15b on IGF1R regulation in vitro are cell line dependent. However, the significant effect of the overexpression of miR-15b on cell proliferation suggest that miR-15b targets other genes related to cell cycle regulation.

### 3.4. Upregulation of miR-15b In Vitro Requires Low Serum and IGF1 Levels in M6 Cells

**miR-15b expression is regulated by low IGF1 levels.** To further investigate the relationship between CR and miR-15b in vitro, we interrogated whether miR-15b is regulated by IGF1 and insulin levels. The M6 and M6C C3-TAg cell lines were treated with 0, 10 or 30 ng/mL of IGF1 (as described in methods). These IGF1 concentrations resemble the levels of IGF1 measured in the serum from CR and control mice, respectively (Figure 1B). In cultured cells, the treatment with 10 ng/mL of IGF1 induced the expression of miR-15b in the M6 cell lines but no in M6C cell lines; however, the addition of 30 ng/mL of IGF1 had not effects on miR-15b expression on both cell lines (Figure 5A). On the other hand, the treatment of M6 cells with different concentrations of insulin had no effects on miR-15b expression (Appendix A). Additionally, the treatment of the M6 and M6C cells with 30 ng/mL of IGF1 increased cell proliferation compared to the cells treated with the lower dose of IGF1 (Figure 5B). These results indicate that IGF1 and not insulin regulate miR-15b expression and suggest a possible negative feedback loop between IGF1 levels and miR-15b expression, in which low IGF1 promote the expression of miR15 while high levels of IGF1 reduce miR-15b expression. Also, these results denote the differential sensitivity of the M6 and M6C cell lines to IGF1 in regulating miR-15b.

**Low serum and IGF1 levels regulate miR-15b expression.** Next, we modified the cell culture media to mimic, in vitro, some of the CR-dependent systemic effects. For these experiments, we used M6 cells because of the high sensitivity to the overexpression of miR-15b and to the levels of IGF1, and we also use M6C, the less responsive cell line. To conduct these experiments, we considered that serum deprivation which reduces hormones, lipids, and growth factors, in combination with glucose reduction, could result in significant cellular stress. To partially mimic CR in vitro, we only reduced the concentration of FBS (see the Discussion, Section 4). First, we tested the expression of miR-15b under different concentrations of FBS (0.5, 1, 2% and 10%). The expression of miR-15b significantly increased only in the cells treated with 0.5% FBS (Appendix A). Then, we cultured M6 and M6C for 18 h in complete media (CM, 10% FBS); CM + 100 nM BMS754807 (a potent and reversible inhibitor of the IGF1R/IR family kinases (Ki, <2 nmol/L)) [56]; 0.5% serum, 0.5% serum + 10 ng/mL IGF1; 0.5% serum + BMS754807; or 0.5% serum + 10 ng/mL IGF1 + 100 nM BMS754807. In both cell lines cultured with CM media (10% FBS) in the presence of IGF1R inhibitor (BMS), no significant changes in the expression of miR-15b or IGF1R were measured. In M6 cells (Figure 5C,D) but not in M6C cells (Figure 5E,F), serum restriction (0.5% FBS) alone elicited an increased expression of miR-15b (Figure 5C) and a decreased expression of *Igf1r* (Figure 5D). Treatment of M6 cells with serum restriction in the presence of 10 ng/mL IGF1 potentiated the expression of miR-15b but did not impact the expression of *Igf1r* relative to serum restriction alone. In media supplemented with 0.5% FBS, the inhibition of IGF1R with BMS754807 had no effect on the expression of miR-15b but it increased *Igf1r* expression, in contrast to what we observed in the cells cultured with complete media and BMS754807. However, the effects of 10 ng/mL IGF1 on miR-15b and *Igf1r* expression were abolished by the addition of BMS754807 to M6 cell cultured in media supplemented with 0.5% FBS and 10 ng/mL of IGF1. Together these results show that in M6 cells, partial serum starvation in combination with low levels of IGF1 can upregulate the levels of miR-15b via the activation of IGF1R. This effect is abolished by the presence of BMS754807, an IGF1R inhibitor, suggesting that in M6 cells, the expression of miR-15b and *Igf1r* requires an active IGF1/IGF1R interaction.

### 3.5. In Silico Analysis Corroborates miR-15b as a Regulator of IGF1/mTOR Signaling

**Identification of target of miR-15.** To identify common target genes for the mature forms of miR-15b in humans and mice, we conducted in silico analysis using a series of webtools for predicting miR-15b targets. Combining the predicted miR-15b target from miRDB, miRabel, miRmap, and TargetScan databases, we identified 14,384 predicted human target genes (hsa-miR-15b-3p and hsa-miR-15b-5p, Appendix A). Moreover, combining predicted targets from miRDB, miRmap, and TargetScan, we predicted 8686 target genes for the mouse (mmu-miR-15b-5p and mmu-miR-15b-3p, Appendix A). Of these genes, 5924 were common between human and mouse (Figure 6A, Appendix A). The common list of miR-15b target genes was compared to a compiled list of 86 genes involved in the IGF1/mTOR pathway. This analysis showed that 52 out of 86 genes were predicted to be a target for miR-15b (Appendix A), suggesting that miR-15b may be involved in the regulation of the IGF1/mTOR pathway.

**Validation of miR-15 targets.** Subsequently, the common list of 5924 target genes was subject to validation in miRTarBase, a database that compiles miR-gene interactions that have published experimental validation. This resulted in a list of 26 validated miR-15b target genes (Figure 6A). The expression of these genes was determined by qPCR in M6 and MDA-MB-231 cells transfected with an miR-15b mimic or with an miR-15b inhibitor. In M6 cells, *Arl2*, *Axin2*, *Ccne1*, and *Igf1r* were downregulated relative to control cells (Figure 6B, Appendix A). In MDA-MB-231 cells expressing miR-15b mimic, most of the miR-15b target genes were downregulated, but only six were significant, including genes encoding suppressors of apoptosis, such as *BCL2* and *BCL2L2*, and genes involved in cell cycle regulation, such as *CCND1*, *CCNE1*, *CHEK1*, and *EIF4A1* (Appendix A). These results suggest that miR-15b has a role in regulating cellular pathways related to cell metabolism and survival.

## 4. Discussion

Calorie restriction consists of a balanced and moderated decrease in the intake of all nutrients. The effects of CR increasing lifespan and ameliorating various diseases, including breast cancer development and progression, have been widely demonstrated in preclinical models and, more recently, in human studies [6,7,57,58]. However, the stringency, psychological and social-behavioral limitations, low compliance among study participants and impaired wound healing [59], make CR challenging to implement in real life. Alternative dietary interventions attempting to mimic CR, including intermittent fasting, time-restricted eating, and macronutrient modulation (fasting–mimicking diet), are increasingly being considered as viable strategies for cancer prevention and as an adjuvant to improve chemotherapy outcomes [60,61]. Although the effects of CR and fasting cannot be seen independent from each other, there is debate about which dietary intervention provides superior effects. A recent study using a breast cancer model in mice compared the effect of CR to cycles of a fasting–mimicking diet [62]. This work showed that daily CR provides greater protection against tumor growth and metastasis. The authors suggested that this could be due in part to the effect of CR on CD8+ and CD4+ immune cells in peripheral tissues. Still, the mechanism underlying the protective effects of CR against breast cancer development and progression are incompletely understood. In this study, (1) we highlight the use of C3-TAg transgenic mice to investigate the impact of CR in a relevant preclinical model of human TNBC and (2) we suggest a role for miR-15b in the antitumorigenic effects of CR.

The mechanism by which the SV40 TAg causes transformation and tumorigenesis is through the binding and inactivation of the key tumor suppressor proteins p53 and Rb [22]. In humans, *TP53* and *RB1* tumor suppressors are frequently mutated in breast cancer, particularly in TNBC [63]. Basal-like TNBCs are characterized as highly aggressive tumors and are frequently associated with poor clinical outcomes compared to other breast cancer subtypes [64,65]. The aggressive nature of basal-like human breast tumors is modeled well by C3-TAg mice, as the majority of female mice on a control diet develop invasive mammary carcinoma before 20 weeks of age, and most require euthanasia by ~6 months of age due to rapid tumor growth [22]. Consistent with published reports, our longest-living control mouse with a tumor was euthanized at seven months of age. In this research, the primary tumors were used as an endpoint; the mice were euthanized when the primary tumor reached 1.2 cm in any direction. To conduct our studies, we focus on the small secondary tumors which reflect early stages in tumorigenesis and/or metastasis. CR increased time to palpable tumor, decreased tumor multiplicity and tumor growth, extended overall survival, and decreased secondary tumor burden. The strong protective effects CR had against C3-TAg tumor development, a notably aggressive model, is a significant finding to add to the growing evidence showing the antitumor effects of calorie restriction; supporting its use for cancer prevention and to improve response to cancer treatment [21,52,62,66,67]. Other approaches to reduce tumor burden in C3-TAg mice, including treatments with cytotoxic chemotherapy drugs (carboplatin, paclitaxel, and polyethylene glycol-tagged liposomal doxorubicin), kinase inhibitors (erlotinib and lapatinib), cMET inhibitors, and dietary supplementation with flavonol, have been unable to significantly extend lifespan or reduce tumor growth [68,69]. One study showed dramatic tumor regression in C3-TAg mice following dual treatment of tumor-bearing mice with the drugs AZD6244 (selumetinib) and BEZ235 (dactolisib), resulting in the dual inhibition of PI3K/mTOR and MEK signaling [70]. Mice treated with AZD/BEZ showed a 94% response rate, with almost half exhibiting a complete response, contributing to a 115% increase in survival compared to untreated animals. The authors noted the potential toxicity of this combined drug-treatment regimen, as several mice had to stop treatment due to severe weight loss.

In our study, CR also had a suppressive effect on tumor progression, measured through the characterization of the grade of mammary lesions throughout a time course that began at 8 weeks of age, when the first observable lesions began to develop, through 20 weeks of age, which is after most mice progressed to invasive carcinoma when on a control diet [22]. We observed that CR largely inhibited the development and progression of mammary lesions relative to the control diet, as the majority of mammary glands in CR mice throughout all time points maintained mostly normal ducts. Only a modest amount of MIN was observed at any time point in CR mice. Our results establish that in C3-TAg mice, tumor cell proliferation and TNBC disease progression are strongly inhibited by CR.

In pathological conditions, such as metabolic syndrome and obesity, an excess of body fat is associated with increased risk of several cancers, including breast cancer [2,3,71]. The underlying mechanism driving the obesity–breast cancer link are not fully understood, but obesity-associated alterations [72], including hyperinsulinemia and increased availability of IGF1, are associated with increased cell proliferation, migration and metastasis, insensitivity to antigrowth signals, induction of angiogenesis, and exacerbating the adipose-tissue-mediated chronic inflammation [73]. Multiple studies have shown that the protective effect of CR in cancer progression are associated with its regulation of IGF1R signal pathway. For example it was shown that restoring the levels of IGF1 reverses the protective effects of CR in a model of bladder cancer [52]. Others have shown that CR inhibits prostate cancer [67,74] and breast cancer progression [21,62] partially through the regulation of the IGF1/IGF1R axis. Similar to our results, these studies showed that CR results in a significant reduction of IGF1 in mice. In humans, meta-analysis of both clinical trials and observational studies revealed no significant effect of CR on the levels of IGF1 [75] unless protein intake was reduced [9,76]. In addition, it is well established that CR reduces insulin levels, increases insulin sensitivity in humans and mice [75,77,78], and exerts a potent anti-inflammatory effect [79]. In this context, deciphering the mechanism associated with CR is significant for developing treatments that can mimic its beneficial effects.

Increasing evidence is showing that the suppressive effects of CR on carcinogenesis are associated with alterations in miR levels [20,21]. In this report, we focus on the impact of CR on IGF1-related miRs. We found miR15b miR-199a-3p, miR-199a-5p, and miR-486 to be significantly upregulated in tumors from CR mice relative to control mice. The miR-199a family was of particular interest because miR-199a-3p directly targets mTOR, one of the key regulation targets of CR, and miR-199a-5p has been reported to target negative regulators of mTOR signaling [80]. The reduced expression of miR-199a-3p was found in an aggressive grade of human breast tumors, and overexpression in MDA-MB-231 cells inhibited the proliferation and the potential to migrate and invade [81]. Expression of miR-199a-5p, on the other hand, was found to be reduced by IGF1 exposure in vitro, and overexpression of miR-199a-5p in MCF7 and MDA-MB-231 cells reduced the activation of AKT in response to IGF1 and inhibited cellular proliferation and invasion, highlighting the potential involvement of this miR in the outcomes of CR. However, although these miRs were highly expressed in tumors, they were not expressed in the C3-TAg progression series of cell lines. These results suggest that miR-199a expression is lost in mammary tissue during tumorigenesis, and that this loss is prevented by CR in our spontaneous C3-TAg tumor model. Alternatively, these miRs that were not expressed in the C3TAg cell lines are not expressed in mammary epithelial cells, and the expression observed in our tumor study was associated with a different cellular component within the heterogeneous tumor environment. There is evidence to support both hypotheses. First, miR-199a, and miR-486 were found to be primarily expressed in normal epithelium, as no expression was found in a wide array of tumors from mammary cancer models, suggesting that these miRs function as tumor suppressors that are lost during tumorigenesis [51]. Another study found miR-199a was expressed in normal breast tissue, particularly fibroblasts, but not expressed at all in proliferating normal or neoplastic cell lines, including the transformed normal breast cell line MCF10A [82]. These findings suggest that miR-199a expression might be confined to the fibroblast-rich stromal compartment in mammary tissue. Because of these prior reports, we interrogated CR’s effect on miR-15b rather than miR-199a or miR-486 expression. In future studies, we plan to further investigate the relationships between CR, miR-199a and/or miR-486 and the tumor microenvironment.

Our current findings suggest a role for miR-15b as a contributor to the anticancer effects of CR. We hypothesized that the antitumor effects of CR are in part mediated by the upregulation of miR-15b via its interactions with *Igf1r*. In this research, we started to delineate a possible mechanism for the antitumor effect of calorie restriction (CR → increases mirR-15b → reduces IGF1R → reduced tumor growth). First, we show in vivo that CR reduces tumor grow, IGF1 levels, and decreases the transcript and protein levels of IGF1R in mammary tumors (Figure 1). Second, we show that miR-15b is significantly expressed in tumors from CR mice and that the expression of miR-15b is affected by tumor aggressivity (Figure 2). Third, using two different methods, our studies showed that increased expression of miR-15b reduces cell proliferation (Figure 3 and Figure 4). Fourth, we show in vitro that miR-15b expression is affected by IGF1 levels (Figure 5A). The expression of miR-15b reduces the transcript levels of IGF1R, and this effect is abolished by an IGF1R inhibitor (Figure 5).

These provocative findings do not establish the causal relationship between CR, miR-15b and mammary tumorigenesis; in particular, additional in vivo studies in combination with an miR-15b inhibitor are necessary to further test a direct mechanism between CR and miR-15b. However, the central findings presented here suggest miR-15b’s function as a possible tumor suppressor through its regulation of *Igf1r*, resulting in a reduction of IGF1R signaling. The degree of miR-15b-dependent reduction in IGF1R seems to be cell dependent. For example, our results show that the overexpression of miR-15b had a profound effect of the IGF1R levels in M6 cells (invasive carcinoma) compared to M6C (metastatic) cells. In M6 and M6C cells, the reduction of IGF1R correlated with decreased levels of p-RPS6, and we also measured a reduction in p-RPS6 in MDA-MB-231 cells. The phosphorylation of RPS6 is a marker for the activation of the IGF1R/PI3K/AKT/mTORC1 pathway associated with resistance to therapy in breast cancer [83]. Our results show a positive correlation between decreased levels of IGF1R, p-RPS6, and a reduction in cell proliferation. These results in the context of CR are consistent with the substantial evidence for RPS6 as a possible anticancer target [84].

The cell-dependent sensitivity to miR-15b effects on IGF1R could be related to the cell culture conditions. We conducted our experiments using 25 mM of glucose in the cell culture media and, to mimic CR, we only reduced the serum concentration. In cell culture, glucose provides the reducing power to neutralize oxidative stress, to maintain the pool of NADPH and to provide energy. We considered that serum deprivation, which reduces hormones, lipids, and growth factors that are essential for cell proliferation and growth, in combination with glucose reduction, could result in significant cellular stress. However, previous studies have shown that the rate of glucose uptake varies among cell types, so it is possible that the less sensitive M6C cells are more efficient at uptaking and metabolizing glucose [85,86]. For future studies in vitro we are considering the optimization of a cell-specific CR medium which requires accurate measurement of the intracellular concentration of glucose to maintain it at physiological levels.

The finding that miR-15b is upregulated in tumors from CR mice relative to control-diet mice is important and is supported by many studies that have clearly demonstrated the actions of miR-15b as a tumor suppressor. For example, it was shown that the expression of miR-15b suppresses the proliferation and invasion of glioblastoma cells by targeting *Igf1r* expression [34]. Other studies using ovarian cancer cell lines showed that the expression of miR-15b represses *WNT7A*, a gene that promotes ovarian cancer cell proliferation [87]. Also, the downregulation of miR-15b has been found in several TNBC tumors and in breast cancer stem cells, and the overexpression of miR-15b inhibits their growth and differentiation [88,89]. It was also shown that the deletion of miR-15b may play a role in B-cell malignancies [90]. However, in other studies, the expression of miR-15b was identified to be upregulated in pancreatic cancer cell lines and tissues, and the overexpression of miR-15b enhanced pancreatic cancer cell invasion, growth, and EMT via regulating the expression of *SMURF2.* Additionally, studies have shown that the expression of miR-15b promote cell growth in MDA-MB-231 [91], and it was downregulated in cells in response to 48 h of serum starvation and 1 μM doxorubicin treatment for 24 h [92]. Together, these reports suggest the impact of miR-15b in cancer may be dependent on tumor-intrinsic characteristics. In our research the overexpression of miR-15b on MDA-MB-231 significantly reduced cell proliferation; this result does not support the findings of Wu et al., 2020 [91]. This could be associated with differences in experimental conditions. For example, in our experiments we performed a transient transfection using 10 nM of miR-15b mimic, while the mentioned study used 50 nM. Future studies are necessary to clarify these differences and for evaluating the involvement of miR-15b in pathways associated with drug sensitivity in the context of obesity, CR, or other physiological conditions.

Various studies have shown that the expression of *Igf1r* has an inverse relationship to miR-15b [33,34]. In this research, we measured a significant miR-15b-dependent regulation of *Igf1r* in M6 cells (carcinoma) but not in the metastatic lines MDA-MB-231 and M6C. This cell dependent effect is expected given that *Igf1r* is often overexpressed in breast cancers [14,15] and analyses using databases such as Oncomir and dbDEMC, which have compiled expression data of miR in tumor and normal tissue from patients, showed that miR-15b is upregulated in invasive carcinoma and in other breast cancer subtypes compared to normal tissue [93]. Our results also showed that low levels of IGF1 induce the expression of miR-15b in M6 cells (carcinoma), and miR-15b inhibition induced the expression of *Igf1r*, suggesting a possible feedback loop between IGF1/IGF1R and miR-15b. The nature of this regulation needs further evaluation. It has been shown that IGF1R translocates to the nucleus following clathrin-mediated endocytosis, regulated by IGF levels [94,95,96]. Thus, future studies should investigate the direct transcriptional regulation of IGF/IGF1R on mir15b using a combination of a translocation blockade along with selective IGF1R inhibitors.

Our results suggest that CR may diminish IGF1/IGF1R signaling in two ways: by reducing circulating ligand, and by repressing *Igf1r* expression in a miR-15b-dependent manner in vivo (mammary tumors). The dual repression of IGF1 signaling evoked by CR is significant as a reduction in ligand alone is difficult to achieve in humans and may not always lead to meaningful pathway suppression and downstream inhibition of tumor growth and progression. Importantly, overexpression of miR-15b, mimicking CR’s effects on this miR, inhibits basal-like mammary cancer cell proliferation, indicating that reduced IGF1R, and possibly the reduction of other miR-15b targets such us *BCL2*, *BCL2L2*, *CCND1*, and *CCNE1*, contribute to the anticancer effects observed with CR in the C3-TAg mouse model and associated in vitro models. Our in silico analysis also showed that miR-15b targets *INSR,* the gene that encodes for the insulin receptor (IR)*,* which suggest that miR-15b targets the IR/IGF1R ratio. Studies have shown that women with high IR/IGF1R ratio have poor breast cancer prognosis [97]. This ratio was elevated in black women compared to white women. Future studies of the impact of CR and miR-15b in breast cancer should consider racial disparities in breast cancer. Another relevant candidate target of miR-15b with one hit in the in silico analysis was *AGER*; this gene encodes for advance glycation end-product receptor (RAGE). A higher expression of RAGE is associated with obesity, inflammation and cancer. Interestingly, a bidirectional regulation between both IR/IGF1R and RAGE has been suggested [73], and, consistently, studies have shown that CR reduces soluble RAGE and advanced glycation end products [98,99]; therefore, further studies will consider investigating the relationship between CR, miR-15b and RAGE in mammary tumors. Expanding on these mechanisms will support our results that miR-15b functions as a tumor suppressor in TNBC, which is in accordance with other reports illustrating the tumor-suppressive, antiproliferative functions of miR-15b in ovarian cancer, glioblastoma cells, and breast cancer stem cells [34,87,88].

## 5. Conclusions

In summary, our findings suggest miR-15b as another component of CR’s regulation of IGF1 signaling that may contribute to the anticancer properties of this dietary intervention. The low levels of circulating bioavailable IGF1 associated with the calorie-restricted state could induce the expression of miR-15b, which further downregulates IGF1 signaling by inhibiting *Igf1r* expression. Importantly, CR-induced miR-15b expression has a significant effect in reducing cancer cell proliferation. We conclude that miR-15b represents a promising new therapeutic target for mimicking some of the anticancer effects of CR.

## Figures and Tables

**Figure 1 cancers-15-04320-f001:**
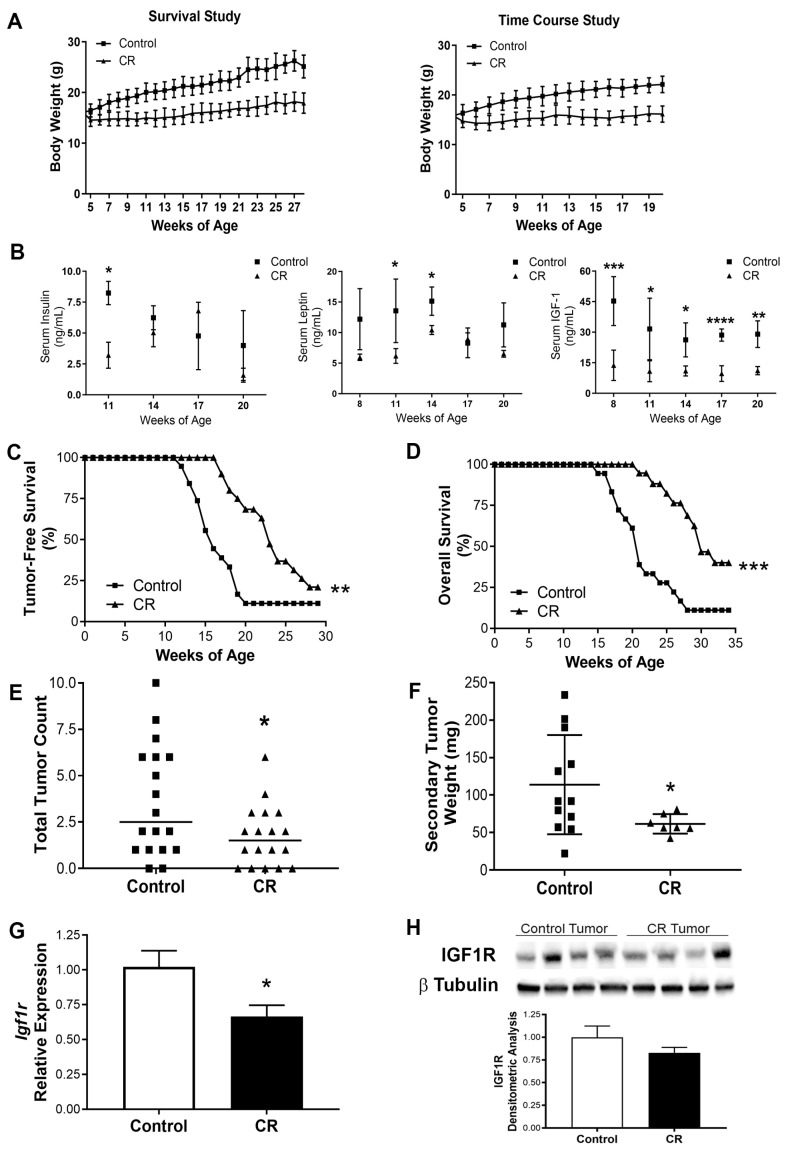
Calorie restriction leads to maintenance of lower body weight, reduces circulating levels of pro-growth hormones, increases survival and decreases tumor burden and *Igf1r* expression in a model of triple-negative mammary cancer. (**A**) Average body weights of mice in survival and time-course studies. Body weights from the survival study were graphed until the week of age at which only one mouse remained in either group. (**B**) Serum hormone concentrations (ng/mL) in mice at corresponding weeks of age of time-course study (control diet *n* = 4–8, CR diet *n* = 3–8). The CR regimen extended (**C**) the time of initial detection of palpable tumors in CR (*n* = 19) relative to control-diet-fed (*n* = 18) C3-TAg mice; and (**D**) the overall tumor-free survival in CR (*n* = 15) versus control-diet-fed (*n* = 17) C3-TAg mice. Consistently, the CR regimen reduced (**E**) the total tumor count; and (**F**) the secondary tumor weights for CR mice relative to control-diet-fed mice at time of euthanasia. (**G**) Expression analysis by qRT-PCR of *Igf1r* in secondary tumors (*n* = 4) from CR relative to control-diet-fed mice. (**H**) IGF1R levels in secondary tumors from CR mice and control-diet-fed mice (*n* = 4) by Western blot, with the corresponding densitometry analysis. Values represent means ± SD; *, *p* < 0.05; **, *p* < 0.01; ***, *p* < 0.001; ****, *p* < 0.0001.

**Figure 2 cancers-15-04320-f002:**
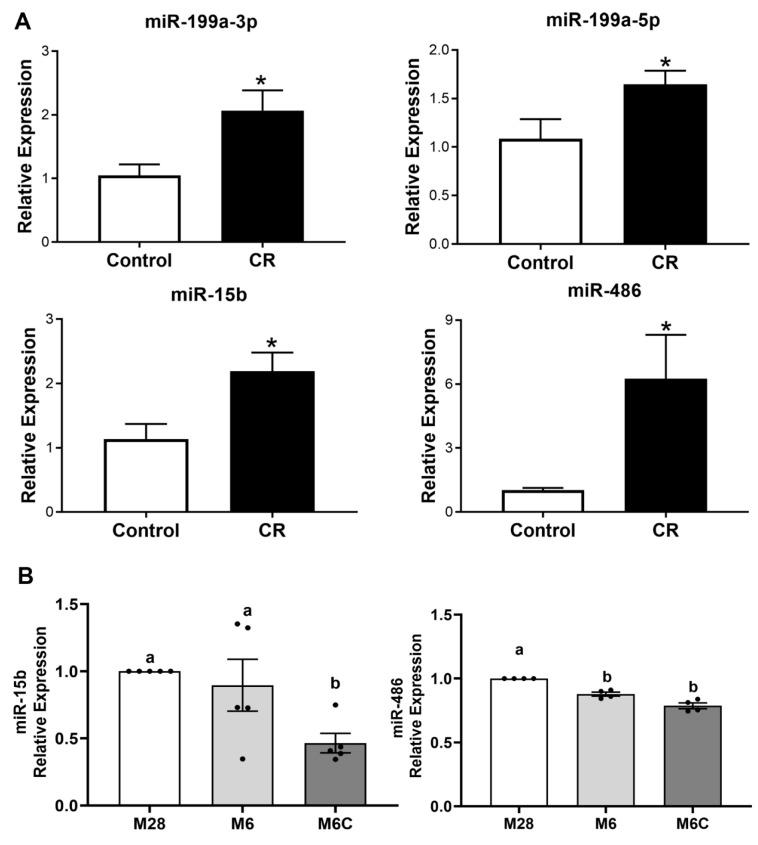
Calorie restriction modulates expression of miRNAs associated with IGF1/mTOR signaling. (**A**) Expression analysis of miRNAs miR-199a-3p, miR199a-5p, miR-15b, and miR-486 in secondary tumors from CR mice (*n* = 7) and control-diet-fed mice (*n* = 5). Values are mean ± SEM; *, *p* < 0.05. (**B**) Expression analysis of miR-15b (*n* = 5) and miR-486 (*n* = 4) in C3-TAg-derived mammary cancer cells M28 (weakly tumorigenic), M6 (invasive carcinoma), and M6C (metastatic) cell lines. Values graphed are mean ± SEM; Values with different letters are statistically different at *p* < 0.05.

**Figure 3 cancers-15-04320-f003:**
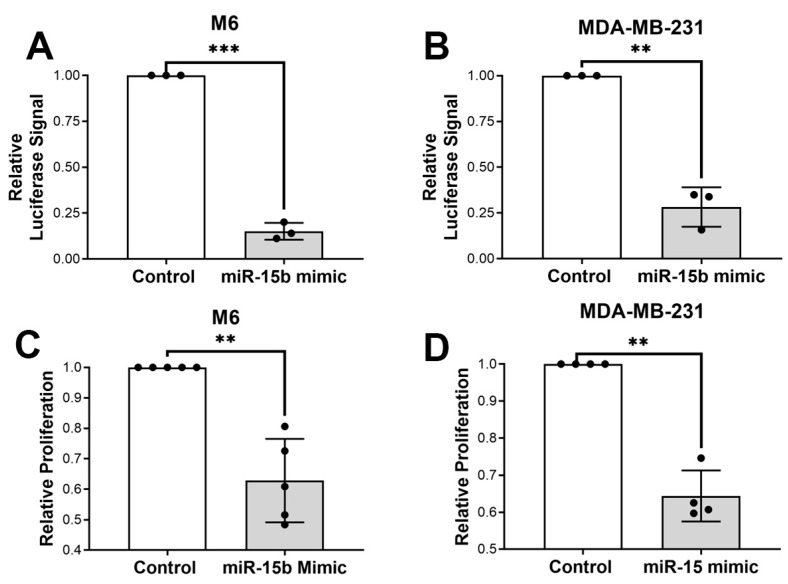
Overexpression of miR-15b reduces cellular proliferation of basal mammary cell lines. (**A**) Murine M6 and (**B**) human MDA-MB-231 basal mammary cell lines were co-transfected with 10 nM of miR-15b mimic or 10 nM of mimic control (control) and a synthetic miR-15b target RenSP luciferase reporter plasmid. The overexpression of miR-15b reduced the luciferase signal evaluated 72 h after transfection in both cell lines (*n* = 3; M6, *** *p* = 0.001; MDA-MB-231 cells, ** *p* = 0.0074). (**C**,**D**) The overexpression of miR-15b reduced cell proliferation in both M6 and MDA-MB-231 cells measured by BrdU incorporation (*n* = 4; M6, ** *p* = 0.004; MDA-MB-231, ** *p* = 0.002).

**Figure 4 cancers-15-04320-f004:**
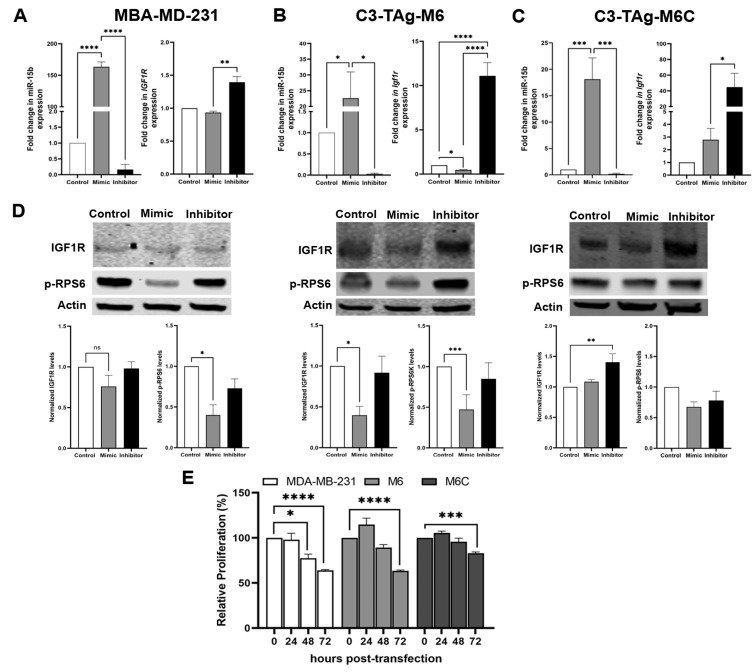
A Transient overexpression of miR-15b with a miR-15b mimic resulted in a cell-dependent effect on *Igf1r* expression 24 h after transfection in (**A**) MBA-MD-231; (**B**) M6 and (**C**) M6C cells 24 h after transfection (*n* = 6). The treatment of the cell lines with an miR-15b inhibitor resulted in the upregulation of *Igf1r* (*n* = 6). (**D**) Representative Western blot for IGF1R and p-RPS6 levels 48 h after transfection with the corresponding densitometry analysis (*n* = 4). (**E**) The overexpression of miR-15b reduces cell proliferation in all cell lines measured by MTT analysis 72 h post-transfection (*n* = 4 each time point). Values graphed are mean ± SEM (*: *p* < 0.05, **: *p* < 0.01, ***: *p* < 0.001, ****: *p* < 0.0001).

**Figure 5 cancers-15-04320-f005:**
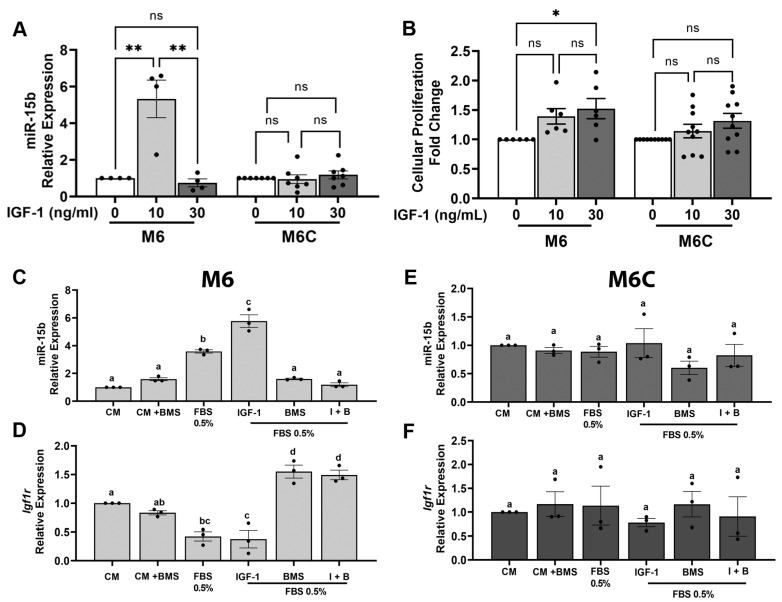
Serum starvation in combination with IGF1 signaling regulators modulates miR-15b and Igf1r expression. M6 and M6C cells were cultured and treated with IGF1 concentrations measured in the serum from CR (10 ng/mL) and control-diet-fed mice (30 ng/mL) to measure (**A**) the relative expression of miR-15b (M6, *n* = 4; M6C, *n* = 7; * = *p* < 0.05; ** = *p* < 0.01) and (**B**) cell proliferation by MTT assay (M6, *n* = 6; M6C, *n* = 10). The expression of miR-15b and *Igf1r* (**D**,**F**) was measured in (**C**,**E**) M6 and (**D**,**F**) M6C cells cultured for 18 h in complete media (CM, 10% FBS), or CM + 100 nM BMS754807 (BMS) (inhibitor of IGF1R), 0.5% serum, 0.5% serum + 10 ng/mL IGF1, 0.5% + 100 nM BMS, and 0.5% serum + 10 ng/mL IGF1 + 100 nM of BMS (I + B) (*n* = 3). Each bar represents the mean + SD. Differences were analyzed using one-way ANOVA, followed by Tukey’s post hoc test. Values with different letters are statistically different at *p* < 0.05.

**Figure 6 cancers-15-04320-f006:**
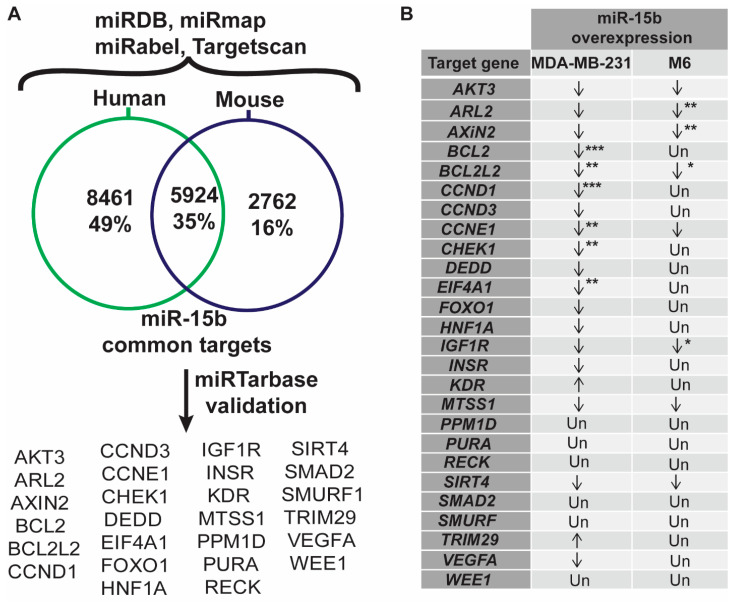
Prediction and validation of miR-15b targets. (**A**) In silico analysis of miR-15b target genes in human and mouse identified 5924 common targets. Of these, 26 genes have experimental validation in miRTarbase. (**B**) Summary of the expression analysis miR-15b predicted target genes in MDA-MB-231 and M6 cells 24 h after treatment with miR-15b mimic. Genes involved in IGF1 signaling, anti-apoptosis and cell cycle were downregulated (*: *p* < 0.05, **: *p* < 0.01, ***: *p* < 0.001; Un: unchanged relative to control).

## Data Availability

The authors confirm that the data supporting the findings of this study are available within the article and/or its Appendix A.

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
