# Peer review of "Regulation of IGF1R by MicroRNA-15b Contributes to the Anticancer Effects of Calorie Restriction in a Murine C3-TAg Model of Triple-Negative Breast Cancer"

_cancers, 2023, doi:10.3390/cancers15174320_

Round 1

Reviewer 1 Report

In the manuscript titled “Regulation of IGF1R by MicroRNA-15b Contributes to the Anticancer Effects of Calorie Restriction in a Murine C3-TAg Model of Triple-Negative Breast Cancer”, the authors showed that calorie restriction’s anti-cancer effect is in part due to the upregulation of microRNA 15b which regulates the IGF1/IGF1R signaling pathway. The study is well designed and the findings are interesting.

Major comments-

1.       The first paragraph of the introduction talks about obesity but the study does not use models of obesity.  This part should either be removed or an obesity model be added to the study.

2.      The CR diet regimen used for mice is faulty. This method of reducing intake by 30% based on intake of control mice is inaccurate as the food intake for each mouse differs. In other articles cited by the authors, the food intake is calculated for each mouse and then gradually reduced by 30-40%.

3.     In the in vitro assay for calorie restriction, cells are cultured 4 hours in serum-free media and are then cultured in media containing 0.5% FBS. During calorie restriction in mice/humans, there is an overall decrease in intake of all nutrients. Please explain the rationale behind decreasing the amount of FBS alone and not glucose.

4.     The reference gene used for PCR-based analyses for microRNA is miR-16. The reference gene should be one that remains unchanged. Several papers show that miR-16 is involved in regulating/ suppressing breast cancer. Would suggest using other small nuclear RNA.

5.     In Fig 1B., do any of the markers show a significant difference between Control and CR?

6.     The whole western blot has not been submitted for Fig 1H.

7.     Figure 2., expression of microRNAs is only in secondary tumors. Why was this experiment not performed using primary tumors?

8.     In Fig 5A, miR-15b expression is increased by 10 ng/mL compared to control but not with 30 ng/mL. How can this be explained? This is again seen in Fig 5C. with IGF-1 in CR media, in FBS 0.5% alone expression if miR-15b is low but the expression was increased with IGF-1 in CR media.

9.       In Fig 1D., treatment with IGF-1 in CR media reduces IGF1R expression but when combined with IGF1R inhibitor BMS in CR media the expression is increased while control CM+BMS the IGF-1R expression is reduced. There is no explanation given for the confusing results.

10.   There is very little discussion on the significance of the findings of this study.

fine

Reviewer 2 Report

This is an interesting study showing how calorie restriction (CR) triggers anticancer effects in Triple negative breast cancers (TNBC) through the regulation of IGF1R. The Authors employ animal models and in vitro models to establish that CR diminishes IGF1/IGF1R signaling. This occurs through miR-15b-mediated repression of IGF1R and reduced circulating levels of IGF1. The experimental design is clear, however few points need to be addressed and/or discussed:

1) The quality of western blotting images in FIg. 4D is extremely poor, so these should be replaced with other blots;

2) Fig. 3C-D: the data on cell proliferation should include the re-expression of IGF1R to assess whether re-expression overcomes the effects of the mimic;

3) Fig. 5A "I" for IGF-1 is missing;

4) Did the Authors verify whether the Insulin Receptor is down-regulated as it seems to be a target of miR-15B (Fig. 6 B); furthermore, CR reduces circulating levels of Insulin, suggesting that some of the effects of CR could be mediated also by alterations of the Insulin/InsR signaling;

5) CR is known to play a relevant role in controlling Ins/IGF pathway and cancer inflammation (10.1210/endrev/bnad005) and it has advantages compared to time-restricted feeding (10.1038/s41467-021-26431-4). These are important points that should be discussed.

Round 2

Reviewer 2 Report

The Authors have addressed all my concerns and the manuscript is highly improved.